# Increased Endotoxin Activity Is Associated with the Risk of Developing Acute-on-Chronic Liver Failure

**DOI:** 10.3390/jcm9051467

**Published:** 2020-05-14

**Authors:** Hiroaki Takaya, Tadashi Namisaki, Shinya Sato, Kosuke Kaji, Yuki Tsuji, Daisuke Kaya, Yukihisa Fujinaga, Yasuhiko Sawada, Naotaka Shimozato, Hideto Kawaratani, Kei Moriya, Takemi Akahane, Akira Mitoro, Hitoshi Yoshiji

**Affiliations:** Department of Gastroenterology, Nara Medical University, Kashihara, Nara 634-8522, Japan; tadashin@naramed-u.ac.jp (T.N.); shinyasato@naramed-u.ac.jp (S.S.); kajik@naramed-u.ac.jp (K.K.); tsujih@naramed-u.ac.jp (Y.T.); kayad@naramed-u.ac.jp (D.K.); fujinaga@naramed-u.ac.jp (Y.F.); yasuhiko@naramed-u.ac.jp (Y.S.); shimozato@naramed-u.ac.jp (N.S.); kawara@naramed-u.ac.jp (H.K.); moriyak@naramed-u.ac.jp (K.M.); stakemi@naramed-u.ac.jp (T.A.); mitoroak@naramed-u.ac.jp (A.M.); yoshijih@naramed-u.ac.jp (H.Y.)

**Keywords:** endotoxin activity, acute-on-chronic liver failure, rifaximin

## Abstract

Acute-on-chronic liver failure (ACLF) leads to systematic inflammatory response syndrome and multiple organ failure. This study investigated the relationship between endotoxin (Et) and ACLF with the aim of determining whether Et activity (EA) is useful as a predictive biomarker of ACLF development and whether rifaximin treatment decreased the risk of ACLF development. Two hundred forty-nine patients with liver cirrhosis were enrolled in this study. Et concentration was determined in the whole blood by a semiquantitative EA assay. Predictive factors of ACLF development and the risk of ACLF development with and without rifaximin treatment were identified by univariate and multivariate analysis using Fine and Gray’s proportional subhazards model. EA level was higher in Child-Pugh class B than in class A patients, and class B patients had an increased risk of ACLF development compared with class A patients. Multivariate analysis showed that EA level was a predictive factor independently associated with ACLF development. Rifaximin decreased EA level and the risk of ACLF development in Child-Pugh class B patients. Et levels were associated with functional liver capacity and were predictive of ACLF development in cirrhotic patients. Rifaximin decreased Et level and the risk of ACLF development in advanced cirrhotic patients.

## 1. Introduction

Acute-on-chronic liver failure (ACLF) develops in cirrhotic patients following bacterial infection, gastrointestinal bleeding, alcohol intake, or worsening of the underlying liver disease [1,2]. ACLF has a high risk of short-term mortality associated with development of multiple organ failure (MOF) [3,4]. The availability of a predictive biomarker of ACLF would improve the prognosis of cirrhotic patients. Increased intestinal permeability in cirrhotic patients may increase the risk of bacterial translocation across the intestinal barrier and inflammation, both of which are involved in the pathophysiology of ACLF [1,2]. Endotoxin (Et), a lipopolysaccharide (LPS), includes a core domain and lipid A in the outer membrane of Gram-negative bacteria [5,6]. LPS that enter the circulation travel to the liver through the portal vein and are recognized by Toll-like receptors that activate innate immune responses including Kupffer and hepatic stellate cells (HSCs) [1,2]. Activated HSCs and Kupffer cells produce nitric oxide and inflammatory cytokines that promote hepatic microcirculatory dysfunction and hepatocyte cell death. The eventual development of systematic inflammatory response syndrome can lead to MOF [1,2]. Et levels are increased in patients with liver cirrhosis and acute liver failure (ALF), and may trigger the development of ACLF [7,8]. The endotoxin activity (EA) assay is a novel alternative to the kinetic turbidimetric Limulus amebocyte lysate test, which is widely used for the detection of Et [6,9]. Previous studies have reported that EA was increased in patients with bacterial sepsis and related to the mortality risk, and that anti-Et treatment decreased EA [9]. Previous studies have also reported that increased EA in cirrhotic patients was associated with progression of liver cirrhosis [10] and EA was increased in patients with ALF [11]. EA associated with bacterial infection and inflammation may be involved in the pathophysiology of ACLF in liver cirrhosis patients. This study investigated the relationship between EA and ACLF, and the usefulness of EA as a predictive biomarker of ACLF development, as well as the relationship between rifaximin treatment and ACLF, and the usefulness of rifaximin treatment as a preventive treatment for ACLF development.

## 2. Experimental Section

### 2.1. Study Design and Patients

This retrospective observational study included a series of 305 cirrhotic patients whose EA was assayed in our hospital between December 2014 and April 2018. Liver cirrhosis was diagnosed by physical findings, laboratory tests, and imaging following the 2015 evidence-based clinical practice guidelines for liver cirrhosis of the Japan Society of Gastroenterology [12]. ACLF was diagnosed by laboratory tests as previously described by Mochida et al. [4]. The description by Mochida et al. included cirrhotic patients with Child-Pugh class A and B and was excluded cirrhotic patients with Child-Pugh class C and uncontrolled hepatocellular carcinoma. Cirrhotic patients with Child-Pugh class C and uncontrolled hepatocellular carcinoma frequently develop liver failure. However, these pathophysiologies were associated with chronic decompensation and cancer and were not ACLF. Because of this, 27 patients with Child-Pugh class C liver disease, 27 with uncontrolled hepatocellular carcinoma according to the description of Mochida et al., and 2 requiring hemodialysis that could affect EA were excluded in our study [6,13]. The remaining 249 patients were included in the analysis. Of 249 patients, 182 were Child-Pugh class A. Of these 182 patients, 8 with Child-Pugh class A were treated with rifaximin for the prevention of hepatic encephalopathy during observation periods and 174 patients were not treated with rifaximin. Further, of 249 patients, 67 were Child-Pugh class B. Of 67 patients, 22 with Child-Pugh class B were treated with rifaximin for the prevention of hepatic encephalopathy during observation periods and 45 patients were not treated with rifaximin. The daily rifaximin dose was 1200 mg in all the patients. All patients on rifaximin treatment were not treated with any other new nonabsorbable disaccharides, probiotics, prebiotics, synbiotics, or antibiotics during observation periods. We initially investigated the relationship between EA and ACLF development in cirrhotic patients with Child-Pugh class A and B. Subsequently, we studied the relationship between rifaximin treatment and ACLF development in cirrhotic patients with Child-Pugh class B (Figure 1). None of the patients had infections including spontaneous bacterial peritonitis or uncontrolled hepatic encephalopathy at the time of assayed EA. The study was approved by the local ethics committee of Nara Medical University (project number: 1453) and was performed in accordance with the ethical standards of the Declaration of Helsinki. Informed consent was obtained from all participants included in the study.

### 2.2. Endotoxin Activity (EA) Assay

Et concentration was determined in whole blood by luminol chemiluminescence using a commercially available semiquantitative EA assay (Spectral Diagnostics Inc, Toronto, Canada (https://spectraldx.com/eaa-for-clinicans) [6,14,15]. The EA assay is based on the binding of Et by an anti-Et antibody with delivery to neutrophils via complement receptors. In the presence of β-glucan and luminol, the neutrophils undergo a respiratory burst accompanied by light emission. The light produced is read by chemiluminometry; the intensity is proportional to the amount of Et. [9]. We divided the patients into low (<0.4) and high (≥0.4) groups based on EA according to a previous study [15].

### 2.3. Statistical Analysis

The statistical analysis was performed using EZR (Saitama Medical Center, Jichi Medical University, Saitama, Japan), which is a graphical user interface for R (version 2.13.0, R Foundation for Statistical Computing, https://www.r-project.org). EZR is a modified version of R commander version 1.6-3 that includes statistical functions frequently used in biostatistics [16]. Results were reported as medians and range. Between-group differences were analyzed using the Mann-Whitney *U*-test. Categorical data were analyzed using Fisher’s exact test. Univariate and multivariate analysis of predictive factors of ACLF development, and the determination of between-group differences in the risk of ACLF development, were performed as described by Fine and Gray’s proportional subhazards model [17]. A two-tailed *p*-value of <0.05 was considered significant.

## 3. Results

### 3.1. Patient Characteristics

The patient characteristics are shown in Table 1. The median age was 69.0 (62.0–76.0) years, 160 were men, 89 were women. Thirty-two had hepatitis B virus, 87 had hepatitis C virus, 73 had abused alcohol, 25 had non-alcoholic steatohepatitis, 12 had primary biliary cholangitis, 7 had autoimmune hepatitis, and 13 had other comorbidities. The observation period after assay of EA was 1099 (range: 802–1163) days. Fifteen patients developed ACLF (2 had Child-Pugh class A and 13 had Child-Pugh class B). Of 15 patients with ACLF development, 9 developed ACLF following bacterial infection, 4 developed ACLF following alcohol intake, and 2 developed ACLF following gastrointestinal bleeding. Thirty patients underwent rifaximin treatment (8 had Child-Pugh class A and 22 had Child-Pugh class B) during the observation period. Aspartate aminotransferase (AST), alanine aminotransferase (ALT), total bilirubin (T-Bil), procollagen-3-peptide (P3P), type IV collagen 7S (4COL7S), and mac-2 binding protein glycosylation isomer (M2BpGi) were higher in patients with ACLF development than in those without (all *p* < 0.05). Albumin (Alb), prothrombin time (PT), platelet count (Plt), and white blood cell (WBC) count were lower in patients with ACLF development than in those without (all *p* < 0.05). Differences in age, sex, etiology, blood urea nitrogen (BUN), creatinine (Cr), ammonia (NH_3_), and C-reactive protein (CRP) were not significant. Patients with ACLF development had a higher risk of esophageal varices and ascites than those without (both *p* < 0.05).

### 3.2. Association of EA with Patient Characteristics

EA was higher in patients who developed ACLF than in those without ACLF (*p* < 0.05, Table 1), and patients were stratified by their EA assay result to low (<0.4) or high (≥0.4) activity groups. Those with high EA had higher AST, T-Bil, and 4COL7S levels and lower Alb and PT levels than those with low EA (all *p* < 0.05, Table 2). EA was higher in patients with Child-Pugh class B disease than in those with class A disease (0.36 (0.32–0.42) versus 0.26 (0.20–0.35), *p* < 0.05). In other words, patients with high EA had significantly worse functional liver capacity than those with low EA. Differences in the age, sex, etiology, ALT, BUN, Cr, NH_3_, Plt, WBC, CRP, P3P, and M2BpGi in patients with high versus low EA were not significant, but ascites was present in more patients with high EA than in those with low EA (*p* < 0.05, Table 2).

### 3.3. Predictive Factors of Acute-on-Chronic Liver Failure (ACLF) Development

Of 249 patients, these analyses excluded the 30 patients (eight had Child-Pugh class A and 22 had Child-Pugh class B) with rifaximin treatment (Figure 1). The univariate and multivariate analysis results are shown in Table 3. Univariate analysis found that Child-Pugh score and EA were associated with ACLF development (all *p* < 0.05) using Child-Pugh score, Model for end-stage liver disease score, Sodium model for end-stage liver disease, and EA [1]. To detect predictive factors for ACLF development, we performed multivariate analysis using Child-Pugh score and EA. These factors had a p-value of <0.05 in the univariate analysis. In the multivariate analysis, EA and Child-Pugh score were independently associated with and predictive of ACLF development. There was no statistical difference between area under receiver operating characteristic (ROC) curve of EA and Child-Pugh score (0.763 versus 0.848, *p* = 0.249). The cumulative incidence of ACLF in patients with low (<0.4) and high (≥0.4) EA are shown in Figure 2a; it indicates that the risk of ACLF development was significantly higher in patients with high EA than in those with low EA (*p* < 0.05). The ROC analysis revealed that a cutoff EA of 0.4 had a specificity of 86.8% and a sensitivity of 35.7%. 

### 3.4. EA and Rifaximin in Child-Pugh Class B

The Child-Pugh class B patient characteristics before rifaximin treatment are shown in Table 4. The observation period with rifaximin treatment group after assay of EA was 1060 (range: 643–1183) days and the observation period without rifaximin treatment group after assay of EA was 952 (range: 381–1106) days. CRP was higher in patients with rifaximin treatment than in those without (*p* < 0.05). However, differences in other parameters, including EA, were not significant. The cumulative incidence of ACLF with and without rifaximin treatment during the observation period (Figure 2b) shows that the risk of ACLF development was significantly decreased by rifaximin (*p* < 0.05). Nineteen patients of 22 Child-Pugh class B patients with rifaximin treatment were assayed for EA before and 30 days after the rifaximin treatment. EA was higher in patients before the rifaximin treatment than in those 30 days after the rifaximin treatment (0.37 (0.32–0.43) versus 0.26 (0.2–0.29), *p* < 0.05).

## 4. Discussion

In this study, Et was predictive of ACLF development in cirrhotic patients. Previous studies reported that progression of liver cirrhosis was associated with intestinal barrier dysfunction leading to several defense mechanism abnormalities and increase in gut-derived endogenous Et [1,2]. Et was found to be associated with functional liver capacity and to increase with the progression of liver disease [10]. In this study, the risk of ACLF development was increased in patients with Child-Pugh class B disease compared with class A disease, and Et was associated with functional liver capacity. Both results support the predictive value of Et as a biomarker for development of ACLF.

In a rat model of endotoxemia, injection of LPS in the mesenteric vein increased vascular permeability and resulted in ascites [18]. In this study, the presence of ascites increased the risk of ACLF development, which is in line with previous findings of increased Et in cirrhotic patients [19]. The finding in this study that the presence of esophageal varices increased the risk of ACLF development is in line with that of a previous report that high Et increased the risk of esophageal varices and that Et increased portal blood pressure by promoting the synthesis of endothelin and nitric oxide in HSCs and endothelial cells [20]. The hemodynamic effects of portal hypertension (PHT) have been shown to induce ACLF development, but transjugular intrahepatic portosystemic shunt stent (TIPS) procedures to treat ascites and esophageal varices [21] can decrease the risk of ACLF development [22]. TIPS has been associated with reduced cytokines including C-X-C motif chemokine ligand (CXCL) 9 and CXCL10 in cirrhotic patients with PHT [22,23,24]. The association of Et with PHT and the development of ascites and esophageal varices suggest that Et is predictive of the development of ACLF. However, the difference in the Et in patients with and without esophageal varices was not significant. This lack of difference may be because most esophageal varices were straight, small-caliber form one varices, and most had been treated by endoscopic injection sclerotherapy or endoscopic variceal ligation. Few patients had moderately enlarged, beady form two varices [25].

Prevention of ACLF development in cirrhotic patients improves prognosis. Rifaximin is used to treat hepatic encephalopathy by selective decontamination of the digestive tract. In this and a previous study, rifaximin decreased Et in patients with advanced cirrhosis [26]. Rifaximin-associated decrease in PHT and Et has been found to improve the prognosis of cirrhotic patients by decreasing the risk of spontaneous bacterial peritonitis and rupture of esophageal varices [27,28]. In this study, rifaximin decreased the risk of ACLF development, possibly for the same reasons. Furthermore, a previous study reported that increased intestinal permeability in cirrhotic patients was associated with defects in intestinal tight junction proteins (TJPs) [29]. Other studies as well as our previous study have reported that rifaximin decreased intestinal permeability via the recovery of TJPs [30,31]. In other words, rifaximin may decrease Et level via the decrease of endotoxin-producing bacteria as well as the recovery of TJPs.

In this study, the fibrosis markers 4COL7S, P3P, and M2BpGi were increased in patients who developed ACLF. 4COL7S and M2BpGi have been shown to increase in ALF patients during HSC activation and extracellular matrix remodeling, and the levels of fibrosis markers have been related to ALF prognosis [32]. Previous studies reported that Et induced the conversion of endothelial cells into activated fibroblasts via transforming growth factor beta secretion [33,34]. In this study, 4COL7S levels were increased in patients a high Et. In other words, Et may well be a prognostic biomarker of ACLF. However, in this study, most patients who developed ACLF died within a short time and Et could not be determined as a prognostic biomarker of ACLF. Investigation of the relationship between Et and fibrosis biomarkers and the prognostic value of Et should continue.

The study has some limitations, including enrolment at a single study center and short observation period. Additionally, the fact that bacterial infections and renal dysfunction that occasionally occur in cirrhotic patients may affect the predictive ability of Et was not considered in this analysis [6,9,13]. In summary, Et was associated with functional liver capacity and was independently associated with the development of ACLF in cirrhotic patients. Rifaximin decreased the risk of ACLF development. The relationships of Et and ACLF warrant further study.

## Figures and Tables

**Figure 1 jcm-09-01467-f001:**
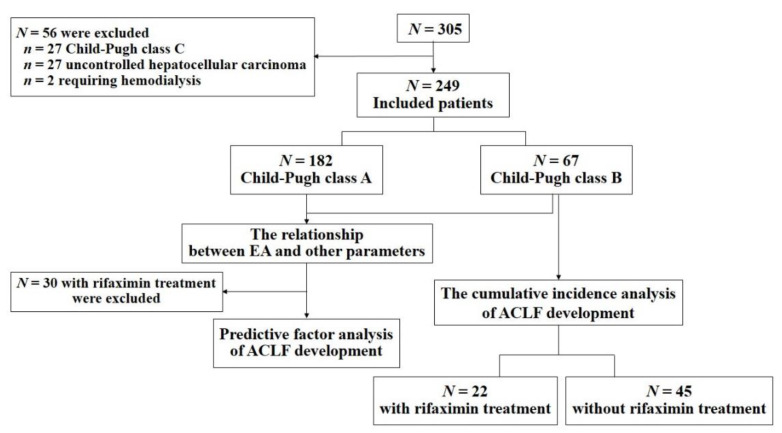
Flow chart of the study. Patients with Child-Pugh class C, uncontrolled hepatocellular carcinoma, and hemodialysis were excluded. The analysis included 249 patients, and 30 out of 249 patients were treated with rifaximin. We initially studied the relationship between endotoxin activity (EA) and other parameters in 249 cirrhotic patients with Child-Pugh class A and B. Subsequently, we investigated the predictive factor of acute-on-chronic liver failure (ACLF) development in 219 cirrhotic patients with Child-Pugh class A and B who were not treated with rifaximin. Sixty-seven patients out of 249 patients were Child-Pugh class B and 22 patients out of 67 patients with Child-Pugh class B were treated with rifaximin. Finally, we studied the relationship between rifaximin treatment and ACLF development in cirrhotic patients with Child-Pugh class B. EA, endotoxin activity; ACLF, acute-on-chronic liver failure.

**Figure 2 jcm-09-01467-f002:**
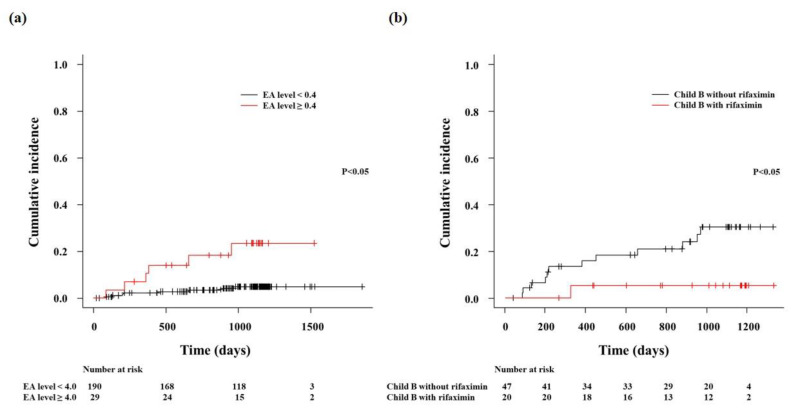
Cumulative incidence of ACLF. (**a**)The risk of ACLF development was significantly higher (*p* < 0.05) in patients with high EA (≥0.4) than in those with low EA (<0.4). (**b**) The risk of ACLF development was significantly lower in Child-Pugh B patients who were treated with rifaximin than in those without treatment (*p* < 0.05). EA, endotoxin activity; ACLF, acute-on-chronic liver failure.

**Table 1 jcm-09-01467-t001:** Characteristics of patients with and without ACLF development.

Variable	Total (*n* = 249)	ACLF Development (*n* = 15)	ACLF Not Development (*n* = 234)	*p* *
Age (year)	69 (62–76)	72 (61–80)	69 (62–75)	NS
Sex (male/Female)	160/89	10/5	150/84	NS
Etiology (HBV/HCV/alcohol/NASH/PBC/AIH/others)	32/87/73/25/12/7/13	0/4/7/2/1/1/0	32/83/66/23/11/6/13	NS
Albumin (g/dL)	4.0 (3.4–4.4)	3.2 (3.0–3.4)	4.0 (3.6–4.4)	<0.05
Aspartate aminotransferase (U/L)	35 (26–47)	65 (42–104)	33 (26–46)	<0.05
Alanine aminotransferase(U/L)	25 (17–37)	40 (25–53)	24 (16–36)	<0.05
Blood urea nitrogen (mg/dl)	15 (12–18)	15 (12–18)	14 (10–18)	NS
Creatinine (mg/dL)	0.83 (0.66–1.01)	0.71 (0.58–0.92)	0.83 (0.66–1.02)	NS
Total bilirubin (mg/dL)	1.1 (0.8–1.4)	1.6 (1.3–2.4)	1.0 (0.8–1.4)	<0.05
Prothrombin time (%)	75 (65–84)	63 (56–71)	76 (67–85)	<0.05
Ammonia (μg/dL)	38.8 (26.2–57.3)	47.8 (38.7–60.6)	37.6 (26.0–56.2)	NS
Platelet count (×10^4^/mm^3^)	10.6 (8.0–14.6)	6.1 (5.4–8.2)	11.1 (8.4–14.9)	<0.05
White blood cell (/μL)	4300 (3200–5500)	3300 (2925–4450)	4350 (3300–5500)	<0.05
C-reactive protein (mg/dL)	0.1 (0.0–0.2)	0.2 (0.1–0.6)	0.1 (0.0–0.2)	NS
Procollagen-3-peptide (ng/mL)	0.9 (0.6–1.1)	1.0 (0.9–1.4)	0.9 (0.6–1.0)	<0.05
Type IV collagen 7S (ng/mL)	7.2 (5.2–10.0)	12.3 (10.9–15.7)	7.0 (5.1–9.7)	<0.05
Mac-2 binding protein glycosylation isomer	2.2 (1.2–5.6)	9.9 (6.4–11.1)	2.1 (1.2–5.4)	<0.05
Esophageal varices (present/absent)	107/142	11/4	96/138	<0.05
Ascites (present/absent)	36/213	6/9	30/204	<0.05
Child-Pugh class A/B	182/67	2/13	180/54	<0.05
Child-Pugh score	5 (5–6)	7 (7–8)	5 (5–6)	<0.05
MELD score	6.4 (3.8–9.3)	6.4 (5.1–9.3)	6.4 (3.7–9.3)	NS
MELD-Na score	7.4 (4.0–10.4)	7.9 (3.3–10.3)	7.4 (4.0–10.4)	NS
EA level	0.29 (0.21–0.36)	0.36 (0.32–0.42)	0.26 (0.20–0.35)	<0.05

Data are expressed as medians (interquartile range); *p* values represent comparisons between cirrhotic patients with ACLF development and not development; ACLF, acute on chronic liver failure; HBV, hepatitis B virus; HCV, hepatitis C virus; NASH, non-alcoholic steatohepatitis; PBC, primary biliary cholangitis; AIH, Autoimmune hepatitis; MELD, Model for end-stage liver disease; MELD-Na, Sodium model for end-stage liver disease; EA, endotoxin activity; NS, not significant; * ACLF development versus ACLF not developed.

**Table 2 jcm-09-01467-t002:** EA level and patient characteristics.

Variable	EA Level < 0.4 (*n* = 219)	EA Level ≥ 0.4 (*n* = 30)	*P* *
Age (year)	70 (62–76)	69 (63–75)	NS
Sex (male/Female)	141/78	11/19	NS
Etiology(HBV/HCV/alcohol/NASH/PBC/AIH/others)	29/75/66/21/9/7/12	3/12/7/4/3/0/1	NS
Albumin (g/dL)	4.0 (3.5–4.4)	3.8 (3.3–4.2)	<0.05
Aspartate aminotransferase (U/L)	33 (26–46)	40 (32–59)	<0.05
Alanine aminotransferase(U/L)	25 (16–35)	27 (20–57)	NS
Blood urea nitrogen (mg/dL)	15 (12–18)	14 (11–19)	NS
Creatinine (mg/dL)	0.83 (0.66–1.0)	0.72 (0.61–0.96)	NS
Total bilirubin (mg/dL)	1.0 (0.8–1.4)	1.3 (0.9–1.6)	<0.05
Prothrombin time (%)	77 (66–86)	69 (63–79)	<0.05
Ammonia (μg/dL)	38.9 (26.2–57.3)	37.6 (26.7–50.7)	NS
Platelet count (×10^4^/mm^3^)	10.4 (8.0–14.6)	11.5 (7.3–14.9)	NS
White blood cell (/μL)	4300 (3300–5500)	4200 (3025–5450)	NS
C-reactive protein (mg/dL)	0.1 (0.0–0.2)	0.1 (0.0–0.3)	NS
Procollagen-3-peptide (ng/mL)	0.9 (0.6–1.0)	0.9 (0.7–1.4)	NS
Type IV collagen 7S (ng/mL)	7.1 (5.2–9.9)	8.6 (6.2–11.9)	<0.05
Mac-2 binding protein glycosylation isomer	2.12 (1.18–5.57)	2.69 (2.00–5.75)	NS
Esophageal varices (present/absent)	96/123	12/18	NS
Ascites (present/absent)	26/193	10/20	<0.05
Child-Pugh class A/B	165/54	17/13	<0.05
Child-Pugh score	5 (5–6)	6 (5–7)	<0.05
MELD score	6.3 (3.8–9.2)	7.4 (3.5–9.7)	NS
MELD-Na score	7.4 (3.9–10.5)	7.8 (6.1–9.0)	NS

Data are expressed as medians (interquartile range); *p* values represent comparisons between cirrhotic patients with EA level < 0.4 and EA level ≥ 0.4; EA, endotoxin activity; HBV, hepatitis B virus; HCV, hepatitis C virus; NASH, non-alcoholic steatohepatitis; PBC, primary biliary cholangitis; AIH, Autoimmune hepatitis; MELD, Model for end- stage liver disease; MELD-Na, Sodium model for end-stage liver disease; NS, not significant; * EA < 0.4 versus EA ≥ 0.4.

**Table 3 jcm-09-01467-t003:** Univariate and multivariate analysis of the association of patient characteristics and development of ACLF.

Univariate Analysis	Univariate Analysis			Multivariate Analysis		
	HR	95%CI	*p*	HR	95%CI	*p*
Child-Pugh score (per 1 increase)	2.289	1.783–2.939	0.0000000009	2.05	1.545–2.722	0.00000067
MELD score (per 1 increase)	1.035	0.963–1.1213	0.35			
MELD-Na score (per 1 increase)	1.031	0.8934–1.19	0.68			
EA level (per 0.1 increase)	974.8	52.88–17970	0.0000037	198.8	5.679–6960	0.0035

ACLF, acute on chronic liver failure; MELD, Model for end- stage liver disease; MELD-Na, Sodium model for end-stage liver disease; EA, endotoxin activity; CI, confidence interval; HR, hazard ratio.

**Table 4 jcm-09-01467-t004:** Characteristics of Child B class patients before rifaximin treatment.

Variable	Total (*n* = 67)	with Rifaximin (*n* = 22)	without Rifaximin (*n* = 45)	*P* *
Age (year)	68 (61–75)	66 (63–70)	71 (58–76)	NS
Sex (male/Female)	42/25	15/7	27/18	NS
Etiology (HBV/HCV/alcohol/NASH/PBC/AIH/others)	2/28/21/5/4/3/4	2/6/7/1/3/1/2	0/22/14/4/1/2/2	NS
Albumin (g/dL)	3.2 (3.0–3.5)	3.2 (2.9–3.4)	3.2 (3.0–3.5)	NS
Aspartate aminotransferase (U/l)	42 (32–63)	35 (31–53)	44 (35–65)	NS
Alanine aminotransferase(U/l)	27 (20–42)	27 (21–44)	25 (17–39)	NS
Blood urea nitrogen (mg/dL)	14 (11–20)	14 (11–17)	14 (12–21)	NS
Creatinine (mg/dL)	0.85 (0.64–1.07)	0.80 (0.61–1.07)	0.86 (0.68–1.07)	NS
Total bilirubin (mg/dL)	1.5 (1.0–2.2)	1.5 (1.0–2.2)	1.5 (1.1–2.3)	NS
Prothrombin time (%)	64 (58–69)	64 (58–69)	64 (59–68)	NS
Ammonia (μg/dL)	51.2 (41.1–72.7)	56.0 (43.0–81.4)	48.8 (39.2–64.9)	NS
Platelet count (×10^4^/mm^3^)	8.4 (5.5–11.3)	8.6 (5.6–9.6)	8.0 (5.4–12.3)	NS
White blood cell (/μL)	3800 (3000–5100)	3600 (3025–4350)	4100 (3000–5300)	NS
C-reactive protein (mg/dL)	0.2 (0.1–0.5)	0.39 (0.1– 0.7)	0.1 (0.0–0.3)	<0.05
Procollagen-3-peptide (ng/mL)	1.1 (0.9–1.4)	0.9 (0.9–1.5)	1.1 (1.0–1.4)	NS
Type IV collagen 7S (ng/mL)	10.8 (9.0–12.7)	9.9 (8.3–12.9)	11.1 (9.9–12.5)	NS
Mac-2 binding protein glycosylation isomer	6.4 (3.4–10.1)	6.4 (3.4–9.7)	6.4 (3.4–10.1)	NS
Esophageal varices (present/absent)	38/29	14/8	24/21	NS
Ascites (present/absent)	25/42	8/14	17/28	NS
MELD score	9.2 (6.1–12.0)	8.0 (6.1–11.6)	9.5 (5.9–12.7)	NS
MELD-Na score	8.0 (3.1–11.2)	4.7 (2.4–11.9)	8.1 (5.3–11.0)	NS
EA level	0.31 (0.23–0.35)	0.32 (0.25–0.37)	0.32 (0.22–0.39)	NS

Data are expressed as medians (interquartile range); *p* values represent comparisons between cirrhotic patients with and without rifaximin treatment; HBV, hepatitis B virus; HCV, hepatitis C virus; NASH, non-alcoholic steatohepatitis; PBC, primary biliary cholangitis; AIH, Autoimmune hepatitis; MELD, Model for end-stage liver disease; MELD-Na, Sodium model for end-stage liver disease; EA, endotoxin activity; NS, not significant; * with rifaximin treatment versus without rifaximin treatment.

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
