# Peer review of "Increased Endotoxin Activity Is Associated with the Risk of Developing Acute-on-Chronic Liver Failure"

_jcm, 2020, doi:10.3390/jcm9051467_

Round 1
Reviewer 1 Report
The article “Increased endotoxin activity is associated with the 3 risk of developing acute-on-chronic liver failure” is well conducted and well written. It include important observation about endotoxin in liver failure. However, in my opinion the discussion section can be improved with intestinal barrier dysfunction mechanism description. Figure 1 and figure 2 should be provide in higher resolution.
Author Response
Response to the Reviewer 1
Thank you for reviewing our manuscript. Our responses to your comments are as follows:
- The discussion section can be improved with intestinal barrier dysfunction mechanism description.
Response to 1)
Thank you for your valuable comment. We have indicated that previous research has reported that increased intestinal permeability in cirrhotic patients was associated with defects in intestinal tight junction proteins (TJPs); further, other studies as well as our previous study have reported that rifaximin decreased intestinal permeability via the recovery of TJPs and that rifaximin may decrease Et level via the decrease of endotoxin-producing bacteria as well as the recovery of TJPs (page 9, lines 247–251).
- Figure 1 and figure 2 should be provide in higher resolution
Response to 2)
We have provided figures of higher resolution to the editorial office.
Reviewer 2 Report
The authors assess the predictive value of endotoxin activity (EA) in development of acute-on-chronic Liver Failure (ACLF) in patients with mild and moderate cirrhosis. In a retrospective study, 249 patients with Child A and B were followed up and the predictor factors for development of ACLF were determined. Child-Pugh score and EA level were found to be independent predictors for ACLF using bivariate Cox regression analysis. The authors also compared two subgroups who had received rifaximin for prevention of hepatic encephalopathy and observed that rifaximin can also reduce the likelihood of ACLF. The study is well designed and investigate a hypothesis that has been tested already in experimental animals. There are many studies to show that injection of endotoxin can cause ACLF is rat models (e.g. doi:10.1088/1361-6579/aaea10). However, clinical studies are essential to show that endotoxin is an important parameter in development of ACLF in patients. The present study uses a sensitive method for measurement of endotoxin in patients suffering from chronic liver failure. I think this study contributes to our current knowledge on pathophysiology and predictors of ACLF. I have the following comments to make some clarifications:
- Why Child-C patients were excluded? The authors have cited a study by Mochida et al. Would it be possible to elaborate more about the reason for excluding Child-C patients?
- Figure 1 (study flow chart) states that Child-A patients who had received rifaximin were excluded from the study. However, in line 118-119 (result section) it is mentioned that “Thirty patients underwent rifaximin treatment (8 had Child–Pugh class A and 22 had 118 Child–Pugh class B) during the observation period”. Please make more clarification. Where this 8 patients with Child-A came from?
- It appears that the authors have used ROC curves to find a cut off value of EA (above 0.4 Or lower than 0.4) to show Cumulative incidence of ACLF (figure 2). Please describe sensitivity and specificity for prediction of ACLF based on this cut off value of EA (0.4).
- In line 22 abstract it s mentioned that “Rifaximin decreased EA level”. This seems to be in contrast with table 4 which shows no significant difference in EA between Rifaximin (+) and Rifaximin (-) groups?
- Why MELD-Na is not measured. It seems to be a better predictor of disease severity in cirrhosis than MELD alone.
Author Response
Response to the Reviewer 2
Thank you for reviewing our manuscript. Our responses to your comments are as follows:
- Why Child-C patients were excluded? The authors have cited a study by Mochida et al. Would it be possible to elaborate more about the reason for excluding Child-C patients?
Response to 1)
Thank you for your valuable comment. We have indicated that cirrhotic patients with Child–Pugh class C and uncontrolled hepatocellular carcinoma frequently develop liver failure; the associated pathophysiologies were chronic decompensation and cancer and not ACLF. Therefore, cirrhotic patients with Child–Pugh class C were excluded (page 2, lines 61–63).
- Figure 1 (study flow chart) states that Child-A patients who had received rifaximin were excluded from the study. However, in line 118-119 (result section) it is mentioned that “Thirty patients underwent rifaximin treatment (8 had Child–Pugh class A and 22 had 118 Child–Pugh class B) during the observation period”. Please make more clarification. Where this 8 patients with Child-A came from?
Response to 2)
We have changed Figure 1 and indicated study design in further detail (page 2, lines 67–69 and page 3, lines 85 and 86).
- It appears that the authors have used ROC curves to find a cut off value of EA (above 0.4 Or lower than 0.4) to show Cumulative incidence of ACLF (figure 2). Please describe sensitivity and specificity for prediction of ACLF based on this cut off value of EA (0.4).
Response to 3)
As per your comment, we have mentioned that the ROC analysis reveals that a cutoff EA of 0.4 has a specificity of 86.8% and a sensitivity of 35.7% (page 6, lines 176–177).
- In line 22 abstract its mentioned that “Rifaximin decreased EA level”. This seems to be in contrast with table 4 which shows no significant difference in EA between Rifaximin (+) and Rifaximin (-) groups?
Response to 4)
We have shown the parameter before rifaximin treatment in Table 4. EA level decreased after rifaximin treatment.
- Why MELD-Na is not measured. It seems to be a better predictor of disease severity in cirrhosis than MELD alone.
Response to 5)
As per your comment, we have shown MELD and MELD-Na scores in our study (Table 1–4).
Round 2
Reviewer 1 Report
The study can be accepeted in present form.
Reviewer 2 Report
The the manuscript has been significantly
improved and now warrants publication in JCM.